# The As and Bs of HIV and Hepatitis Co-Infection

**DOI:** 10.3390/tropicalmed4020055

**Published:** 2019-03-27

**Authors:** Darcy Wooten, Maile Y. Karris

**Affiliations:** Division of Infectious Diseases and Global Public Health, University of California, San Diego, CA 92103, USA; m1young@ucsd.edu

**Keywords:** hepatitis A, hepatitis B, HIV, co-infection

## Abstract

Hepatitis A and B co-infection among people living with HIV are public health challenges that account for an increasing degree of morbidity and mortality. Understanding the changing epidemiology, clinical manifestations, and new approaches to treatment and prevention continues to be important in the care of people living with HIV. We conducted a review of the literature that included studies on hepatitis A and HIV co-infection and hepatitis B and HIV co-infection, focusing on epidemiology, clinical manifestations, treatment, and prevention. Important updates include the changing epidemiology of hepatitis A outbreaks among the homeless and individuals who use substances, and novel approaches to hepatitis B vaccination and hepatitis B cure strategies.

## 1. Introduction

Hepatitis A (HAV) and hepatitis B (HBV) co-infection among people living with HIV (PLWH) contributes to a significant amount of morbidity and mortality worldwide. The changing demographics of HAV outbreaks among vulnerable populations including homeless individuals, people who use drugs, and MSM (men who have sex with men) have important implications for vaccination strategies. HBV and HIV co-infection currently requires lifelong treatment with nucleoside/nucleotide reverse transcriptase inhibitors; however, research to develop either a functional cure or viral eradication is underway. HBV vaccination, resulting in protective seroconversion, remains suboptimal in people with HIV; however, new strategies, including the use of vaccine adjuvants with toll-like receptor agonists, hold great promise. In this review, we provide a general overview of the current literature surrounding HAV and HBV co-infection with HIV.

## 2. HAV and HIV Co-Infection

### 2.1. Epidemiology of HAV and HIV Co-Infection

HAV infection occurs worldwide, with an estimated 1.4 million cases per year [1]. Transmission occurs by the fecal–oral route, either through direct person-to-person contact or exposure to contaminated food or water. HAV is endemic to developing countries with poor sanitation, whereas sporadic, foodborne outbreaks have historically been seen in developed countries [1]. Recently, developed countries are experiencing outbreaks of HAV within specific populations facilitated by direct person-to-person spread of the virus; these groups include homeless individuals, people who use drugs, and men who have sex with men (MSM) [1,2,3,4].

The seroprevalence of HAV infection tends to be higher among PLWH compared to those without HIV (15.1%–96.3%) but varies geographically [1]. The proportion of PLWH who are seropositive for HAV is greatest in countries with low HAV endemicity and is associated with oral-anal sex, number of sexual partners, older age, and injection drug use.

Since the 1980s, epidemiologists have described HAV outbreaks amongst MSM, due to oral-anal and digital-rectal sexual transmission [1,4,5]. In developed countries, due to vaccinations HAV cases have overall decreased including in MSM [6]. However, several developed countries are reporting recent large-scale outbreaks specifically in MSM and PLWH. In 22 countries across Europe, 4475 cases of HAV have occurred since 2016 (in Austria Belgium, Croatia, the Czech Republic, Denmark, Estonia, Finland, France, Germany, Greece, Ireland, Italy, Latvia, Luxembourg, Malta, the Netherlands, Norway, Portugal, Slovenia, Spain, Sweden, and the United Kingdom), representing a four-fold increase from baseline [7,8,9,10,11,12,13]. Over 80% of these cases identified as MSM. Similarly, MSM were overly represented (70%) in a 1000-person outbreak in Taiwan in 2016. Of these, 60% were PLWH [4].

The United States is not immune to this worldwide trend, but recent outbreaks of HAV have primarily occured in homeless populations and among people who use drugs [3]. The most recent outbreak originated in San Diego, California, in 2016, but has spread to Kentucky, West Virginia, Ohio, Indiana, and Missouri. Over 15,000 cases are linked in this outbreak and an estimated 140 affected persons have died.

Transmission of HAV was due to unsanitary living conditions among individuals living in tent cities clustered in downtown metropolitan areas. Unlike outbreaks in other parts of the world, PLWH and MSM did not appear to be at higher risk [3]. An outbreak of HAV caused by a different strain of the virus also occurred in Michigan in 2017 among a similar at-risk population of homeless individuals and people reporting drug use.

The outbreaks in Europe, Taiwan, and United States highlight important changes in the epidemiology of HAV infection in developed countries. While foodborne outbreaks will continue to play a role in HAV outbreaks in the future, direct person-to-person transmission in high-risk groups, including MSM, homeless individuals, and people who report drug use, is increasingly important and carries significant implications regarding vaccination recommendations and future public health interventions.

### 2.2. Clinical Manifestations of HAV Co-Infection among People with HIV

Following exposure, the incubation period of HAV is approximately 28 days (range 15–50) [14]. Symptoms in adults are non-specific and include fever, fatigue, malaise, anorexia, abdominal pain, nausea, vomiting, diarrhea, and jaundice. Symptoms coincide with a rise in the liver function tests, with transaminases rising to 5–10 times the upper limit of normal [15]. Total bilirubin and alkaline phosphatase can also be elevated, although they are typically <10 mg/dL and 400 U/L, respectively. Symptoms and lab abnormalities are self-limited and typically require only supportive care. Over 85% of patients have complete clinical and biochemical recovery within three months of symptoms, and nearly 100% of patients resolve signs and symptoms of infection by six months [15]. Fulminant hepatic failure (characterized by encephalopathy, coagulopathy, and other signs of impaired synthetic liver function) is exceedingly rare; however, the risk increases with age >50 years and with an underlying liver disease, such as cirrhosis or co-infection with HBV or HCV [16].

Complications of HAV are also rare and include cholestatic hepatitis (prolonged jaundice and cholestasis of over three months), relapsing hepatitis (relapse of symptoms and biochemical abnormalities within six months of the initial presentation), and autoimmune hepatitis (characterized by hyperglobulinemia and the presence of autoantibodies) [15,17].

Data suggests that there are very little differences in the clinical presentation of HAV between people with and without HIV co-infection. One study of acute HAV in PLWH and HIV-uninfected persons found that the frequency and severity of symptoms did not differ between the two groups. Interestingly, the degree of transaminase elevation was lower in patients with HIV (specifically those with a detectable viral load and more advanced immunosuppression) [18]. This is presumably due to a blunted immune response to HAV-infected hepatocytes with HIV-associated immunosuppression. A large study in Taiwan also found that people with HIV had lower peak alanine aminotransferase levels, less coagulopathy, and less hepatomegaly or splenomegaly on imaging [19].

Although symptoms and disease severity are similar between these groups, PLWH may demonstrate a slower resolution of their acute HAV and a prolonged duration of HAV viremia. One study demonstrated that PLWH had higher plasma HAV virus loads and a significantly longer duration of HAV viremia (53 vs. 22 days) compared to those without HIV [18]. Additionally, PLWH have prolonged shedding of HAV in the stool. These findings have implications for the duration of clinical monitoring and the duration of transmission risk in patients with HAV/HIV co-infection; however, more data are needed to better inform clinical practice.

### 2.3. HAV Vaccination among People with HIV Infection

Universal vaccination for HAV amongst PLWH is not currently recommended by any of the major society guidelines. A risk-based approach is used, based on the likelihood of exposure (MSM, drug use, travel, occupational exposure, household contacts, homelessness, receipt of clotting-factor products, residents of care institutions, outbreak setting) or the likelihood of more severe disease if exposed (chronic liver disease, immunosuppressed, active HBV or HCV infection) [20]. In several developed countries, including the United States, universal vaccination in childhood is the standard of care [21].

There are two inactivated HAV vaccines and one inactivated HAV/HBV vaccine licensed for patients with HIV. Because of the generally higher prevalence of HAV among PLWH, most society guidelines recommend checking a HAV IgG serology at baseline to determine immune status and vaccination needs.

Seroconversion rates (defined as the development of an IgG titer over 20 mIU/mL) are lower amongst PLWH as compared to HIV-uninfected individuals (48.6%–95% in patients with HIV vs. 97%–100% in patients without HIV) [1,5]. Seroconversion is substantially lower after one dose of vaccine amongst PLWH, compared to those without HIV; following the second dose, seroconversion rates improve dramatically, highlighting the importance of ensuring that PLWH receive both doses of the HAV vaccine. Risk factors for lack of seroconversion include low CD4 T-cell count, high HIV viral load, HCV co-infection, and tobacco use. Studies looking at whether three doses (at week 0, week 4, and week 24) of vaccine could increase rates of seroconversion as compared to the two-dose schedule showed a trend toward higher rates of seroconversion (84.2% vs. 78.1%) but did not reach statistical significance [1,5]. The three-dose schedule does increase the absolute HAV IgG titer and prolongs the duration of protective antibody as compared to two doses. Durability of seroprotection is thought to be shorter amongst PLWH, although data are limited. One study found that five years after vaccination, 76.4% of PLWH remained seroprotected against HAV, whereas 78.9% of HIV-uninfected patients remained seroprotected, although this difference was not statistically significant [1]. PLWH who were on the three-dose (vs. the two-dose) schedule were significantly more likely to remain seroprotective five years after vaccination, as were those with a higher CD4 T-cell count and suppressed viral load at the time of vaccination, and those without hepatitis C co-infection. Yet, the clinical significance of seroreversion is unclear as these individuals may still be protected against HAV if they are re-exposed vis-à-vis memory T-cell immunity.

Society guidelines differ in their recommendations for serologic monitoring and vaccine boosters following primary vaccination. The British HIV Association (BHIVA) recommends an HAV booster every 10 years for HIV-infected patients with ongoing exposure risk (expert opinion) [22,23]. The European guidelines (EACS) recommend the consideration of checking HAV serologies periodically to identify seroreverters to determine if a booster is necessary [23]. The ACIP, Department and Health and Human Services (DHSS), and World Health Organization (WHO) guidelines do not comment on the utility of monitoring post-vaccination serologies or the use of HAV boosters in those identified as not seroprotected [20,21,24,25]. Additional data are needed to guide policies on HAV vaccination recommendations and monitoring for PLWH.

## 3. HBV and HIV Co-Infection

### 3.1. Epidemiology of HBV and HIV Co-Infection

Over 240 million people worldwide are chronically infected with HBV [26], and HBV co-infection is a leading cause of liver disease in patients with HIV. In areas of low HBV endemicity, approximately 7%–10% of patients with HIV are co-infected with HBV [27]. Transmission in this group typically occurs through sexual exposure or injection drug use in adulthood. In contrast, in places of high HBV endemicity, such as Asia and Africa, 20%–30% of patients with HIV are co-infected with HBV. Transmission in these areas occurs perinatally or in early childhood, long before HIV acquisition. The risk of developing chronic HBV infection following exposure is directly related to the age at which an individual is exposed to the virus. Perinatal exposure results in a >90% chance of developing chronic disease whereas <5% of those infected later in life will develop chronic infection [27]. Having HIV at the time of HBV exposure increases the risk of chronic HBV infection, and this risk is increased with lower CD4 T-cell count, higher HIV viral load, and poor overall health [28,29,30]. HIV-HBV co-infection results in significantly accelerated fibrosis and higher rates of cirrhosis, decompensated liver disease, and hepatocellular carcinoma (HCC) compared to HBV monoinfection. Co-infected persons also demonstrate a four-fold higher risk of HCC, a 1.5-fold increased risk of all-cause mortality, and a three-fold increased risk of liver-related mortality compared to HIV-monoinfected persons [29,30]. Baseline screening for HBV co-infection in PLWH, staging of liver disease and fibrosis, and HCC screening as indicated are critically important in treating and preventing disease in those at risk.

### 3.2. Staging Patients with HBV/HIV Co-Infection

All PLWH should undergo HBV testing with an HBV surface antibody (anti-HBs), HBV surface antigen (HBsAg), and HBV core total antibody (anti-HBc) at entry into care [24]. Those who have a persistent HBsAg on two separate occasions six months apart are chronically infected and should undergo additional testing, including an HBV “e” antigen (HBeAg), HBV “e” antibody (anti-HBe), and HBV DNA PCR to determine their stage of chronic infection. PLWH should also be evaluated for HCV and for immunity to HAV. Those with HCV should be treated, and those without immunity to HAV should be vaccinated. There is increasing evidence for the utility of noninvasive methods to evaluate the degree of liver fibrosis, including transient elastography and serum markers [29]. Liver biopsy is rarely necessary and should be individualized based on the clinical scenario.

The significance of an isolated anti-HBc is unclear and, in most cases, signifies past exposure to HBV with subsequent clearance of the virus and loss of anti-HBs [24,29,30]. Alternatively, this could signify chronic infection or, in low-prevalence settings, a false positive test. Testing for HBV DNA is recommended when isolated anti-HBc is found and there is a high pre-test probability for chronic HBV infection.

### 3.3. Treatment of HBV/HIV Co-Infection

All PLWH should be started on antiretroviral (ARV) therapy. For patients co-infected with HBV, a tenofovir-based ARV regimen is recommended. Currently, the most common options for therapy include tenofovir disoproxil fumarate (TDF) or tenofovir alafenamide (TAF) with emtricitabine (FTC) or lamivudine (3TC) [28,29]. Individual factors to consider prior to the initiation of therapy include a history of HBV treatment, especially with non-tenofovir monotherapy (i.e., 3TC, adefovir, entecavir), creatinine clearance, urinalysis, and bone-health history. For the majority of persons with a CrCl >30 mL/min, TAF/FTC as the backbone of the ARV regimen is preferred; TDF/FTC is acceptable for PLWH with a CrCl >60 mL/min in circumstances when TAF is not available. In rare cases, when TDF or TAF cannot be used, entecavir should be added to a fully suppressive ARV regimen. HBV monotherapy with lamivudine (or emtricitabine) without tenofovir is never recommended due to the inevitable development of resistance [29].

The goal of treatment is suppression of HBV DNA, which occurs in the majority of persons within 1–3 years of treatment initiation. Transaminases and HBV DNA should be monitored every 3–6 months to evaluate for response to treatment and to ensure that resistance is not developing. A small proportion of co-infected patients (5%–10%) do not achieve suppression on a tenofovir-based regimen after 3–5 years [24]. Data suggest that noncompliance accounts for the majority of these cases. Whether entecavir intensification is beneficial in these cases remains unknown at this time. Lifelong treatment is recommended, and discontinuation of therapy can be associated with a flare in up to 30% of patients, most commonly in those with underlying cirrhosis [26,30]. Although not recommended, if therapy must be stopped, HBV DNA PCR and transaminases should be checked every six weeks for 3–6 months to monitor for a flare of underlying liver disease.

### 3.4. HBV Vaccination among People with HIV Infection

The initial evaluation for HBV exposure and infection is shown in Figure 1. All PLWH should be tested for immunity and exposure to HBV as outlined above. Non-immune, non-exposed persons (HBsAg negative, anti-HBc negative) should be vaccinated. Those who are anti-HBs positive and anti-HBc positive have resolved infection and do not need to be vaccinated. Those who are anti-HBs positive with a titer of at least 10 mIU/mL, following a complete vaccination series, are protected and do not require vaccination. Individuals with an isolated anti-HBc that is positive should be given a single dose of vaccine; if the anti-HBs titer is greater than 100 mIU/mL, one to two months later, the individual is immune and does not require further vaccination. If the titer is <100 mIU/mL, the individual should complete a standard vaccination series [24].

In the United States, there are four approved HBV vaccines (Table 1). Two are recombinant HBV-surface antigen (rHBsAg) vaccines (Energix-B and Recombivax–HB) dosed at 0, 1, and 6 months, and one is a combined rHBsAg vaccine with an inactivated HAV vaccine (Twinrix). Recently, the Federal Drug Administration approved an rHBsAg vaccine conjugated to a toll-like receptor (TLR)-9 agonist given in two doses (Heplisav B) [31,32]. TLR9 is a sensing receptor for innate immune responses located on dendritic cells and memory B cells. Combining TLR adjuvants with rHBsAg leads to enhanced T- and B-cell memory.

In persons without HIV, the rate of seroconversion to protective levels of surface antibody (defined at 10 mIU/mL) is roughly 96% [33]. PLWH have a significantly lower rate of seroconversion following vaccination (range 35%–70%). Predictors that decrease the likelihood of protective seroconversion include low CD4 T-cell count, detectable HIV viral load, infection, and co-infection with HCV [26,29]. To increase the rate of protective seroconversion, various strategies of primary vaccination and revaccination have been explored, including higher vaccine dose, increase in vaccine-dosing frequency, alteration in dosing site, and addition of adjuvants.

A large study in France that looked at primary HBV vaccination in non-immune PLWH showed that protective seroconversion was more likely and durable with a double dose at 0, 4, 8, and 24 months compared to the standard, single dose given at 0, 4, and 24 months [34]. A similar study in Thailand, comparing a standard dose and schedule to standard dose given at 4 times points and double dose at four time points, did not show different rates of seroconversion or durability [35].

The use of toll-like receptor agonists, in conjunction with rHBsAg, does appear to increase vaccine efficacy in PLWH. A European study showed that a TLR-4 agonist combined with rHBsAg at 0, 1, 2, and 6 months was associated with 100% seroconversion in a small group of PLWH [36]. Revaccination, using a vaccine with rHBsAg plus a TLR agonist, demonstrated greater efficacy compared to revaccination strategies without the TLR adjuvant [36].

For those who do not respond to an initial HBV vaccination series (e.g., post-vaccine titer <10 IU/mL), there are several options, although revaccination data remain limited. Various strategies include repeating a three-dose series at double the dose (40 mcg) or giving a four-dose series (at 0, 1, 2, and 6 months) at either standard dose (20 mcg) or double dose (40 mcg) [24]. Providers may also opt to defer repeating vaccination until the CD4 T-cell count is greater than 200–350 since this increased likelihood of seroconversion.

However, with the arrival of Heplisav B that only requires two doses, these strategies may be unnecessary. In three pivotal registration trials of HIV-uninfected individuals, Heplisav B had superior rates of seroconversion compared to Energix B [31,32,37]. Although studies of Heplisav B have not included PLWH, studies have looked at vaccination efficacy in other immunocompromised recipients, such as persons with diabetes and persons on dialysis and demonstrated improved response rates [38]. Initial concerns existed about an association between Heplisav B with a small increase in ischemic cardiovascular events; however, on further review, this finding was not statistically significant. Heplisav B is currently being studied in PLWH (A5379), with close attention to possible vaccine-associated cardiovascular adverse events given the higher risk of cardiovascular disease at baseline in this population.

### 3.5. New HBV Treatment and Cure Strategies

Newer research is aimed at novel approaches to treating HBV, including cure strategies. Similar to HIV, chronic HBV is challenging to cure because of its stable nucleic acid, covalently closed curricular DNA (cccDNA), which persists in the host nucleus. One advantage to HBV cure strategies over HIV is that the reservoir, hepatocytes, are well defined and finite. There are two proposed approaches to cure: (A) a functional cure in which there is a loss of HBsAg and acquisition of anti-HBs off medications despite persistence of cccDNA and (B) eradication, in which all cccDNA is eliminated from hepatocytes [39]. The two main strategies for cure include virologic approaches and immune approaches. Virologic approaches currently under investigation, and in early phase studies, include therapies that block various stages of the HBV replication cycle, including blocking entry, eliminating or silencing cccDNA, inhibiting capsid formation, or blocking secretion of HBsAg to prevent immune fatigue. Immune approaches include inhibiting immunoregulatory pathways and boosting HBV specific immunity [39]. Based on these studies as well as the experience from HIV- and HCV-treatment development, it is likely that an HBV cure will require combination therapy, targeting multiple steps within the viral life cycle in conjunction with therapies targeting various aspect of the immune response.

## 4. Conclusions

Challenges in the prevention and treatment of HAV and HBV co-infection amongst PLWH persist. With regards to HAV, the changing epidemiology of HAV outbreaks highlight the importance of evaluating whether PLWH are immune to HAV when they enter care, and if not, vaccinating them. Development of societal infrastructures that address the underlying root causes of these HAV outbreaks, including poverty, homelessness, and substance use are critical. One of the biggest areas of uncertainty is the durability of HAV vaccination amongst PLWH, and additional data are needed to determine whether periodic monitoring of serologies and/or a booster vaccination beyond the current HAV two-dose vaccine series would be beneficial. With regards to HBV, all PLWH should be screened for evidence chronic infection upon entry into care and, if found to be chronically infected, initiated on treatment with lifelong tenofovir-based, fully suppressive antiretroviral therapy. In cases where tenofovir-based therapy cannot be used, entecavir can be added to a fully suppressive antiretroviral regimen. Ongoing research focusing on virologic and immunologic strategies to eliminate cccDNA and, thereby, to cure HBV is needed, especially for PLWH, since this group has worse liver-related outcomes as compared with patients with monoinfection. New HBV vaccination strategies that utilize adjuvants, such as toll-like receptor agonists, are likely to be more effective than the original recombinant-surface antigen HBV vaccines in PLWH.

## Figures and Tables

**Figure 1 tropicalmed-04-00055-f001:**
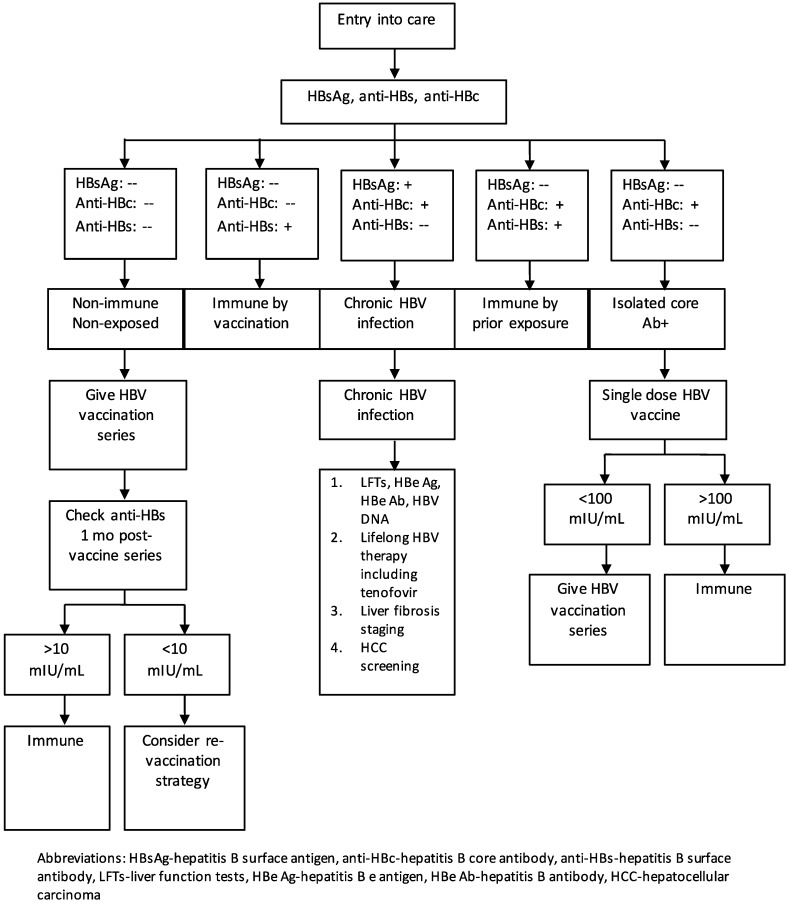
Algorithm of the evaluation and prevention and treatment strategies for hepatitis B virus infection in people living with HIV.

**Table 1 tropicalmed-04-00055-t001:** Available HBV vaccines in the United States.

Brand	Dose(s)	Schedule
Recombivax HB	10 mcg rHBsAg, 40 mcg rHBsAg	0, 1, and 6 months
Energix-B	20 mcg rHBsAg	0, 1, and 6 months
Twinrix	20 mcg rHBsAg + inactivated HAV vaccine	0, 1, and 6 months
Heplisav-B	20 mcg rHBsAg + TLR9 agonist	0 and 1 month

HBV = Hepatitis B virus; rHBsAg = recombinant hepatitis B surface antigen, HAV = hepatitis A virus, TLR9 = toll-like receptor.

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
