# Peer review of "The As and Bs of HIV and Hepatitis Co-Infection"

_tropicalmed, 2019, doi:10.3390/tropicalmed4020055_

Round 1
Reviewer 1 Report
Review report
Title: The As and Bs of HIV and Hepatitis Co-Infection
Authors: Darcy Wooten and Maile Y. Karris. UCSD. USA
Journal: Tropical Medicine and Infectious Disease
Summary:
In this manuscript, the authors summarize the epidemiology, clinical manifestations, treatment and prevention of HIV/HAV and HIV/HBV co-infections. Besides a broad description of these items, the authors also provide updated evidence and make special mention to present HAV outbreaks, especially across the US in populations like the homeless and individuals who use drugs, and to newer strategies for HBV vaccination and cure (functional and microbiological). Present challenges for prevention and treatment of HAV and HBV are described throughout the article. A diagnostic and management algorithm for HBV is also provided.
Broad comments:
The manuscript is well organized and includes all the major components of a review article. Each specific section describes the importance of HAV and HBV co-infection in PLWH. The article is well referenced and up to date and describes an ongoing outbreak of HAV in the US, as well as novel strategies for HBV vaccination and cure. It is easy to understand and adds value to the current knowledge of both co-infections. I don’t see major flaws in the preparation of the manuscript and think it will target an audience composed by providers that take care of PLWH as well as clinicians in training.
I would expand on the conclusions at the end of the paper by briefly summarizing the salient aspects of the disease, but also commenting on the importance of the topic and the future management of the hepatitis-HIV co-infections.
Specific comments:
I will outline my comments in reference to the line numbers, tables and figures as below:
24, 40: use “people who use drugs” instead of “people with drug use”
24: Add “(MSM)” in front of “men who have sex with men”
27 and beyond: use “HBV” instead of hepatitis B”, as it is already defined in line 21
40, 54: make sure the reference on outbreaks is up to date as new information is being reported
43: “endemicity” instead of “endemnicity”
52: “occurred in” instead of “identified as”
91-94: interesting. Consider adding possible explanations for this, e.g. due to blunted immune response, especially if mentioned by authors in references
113: add reference to this statement
145: “240 million people worldwide are infected with HBV”. Add “chronically” if appropriate
165 and beyond: nomenclature for HBV antigens and antibodies should be used once spelled out for the first time. Suggest using standard, well accepted nomenclature: HBsAg, HBeAg, anti-HBs, anti-HBc, etc.
177: Consider adding “Testing for HBV DNA is recommended when isolated anti-HBc is found”. May apply to paragraph 209-213
197: “HBV antigen”: which one? DNA? Please clarify
200: consider writing this subtitle as in HAV section, line 102 (i.e. choose “vaccination” or “prevention” for both to give consistency)
207: Figure: Box under Chronic HBV infection not clear. Item 3 seems to be missing information: “liver fibrosis…”
207: Figure: suggest nomenclature as above
242-243: add “rHBsAg”
263: define cccDNA
277: suggest adding much more text to the conclusions as I don’t think that the manuscript is about challenges as described by the authors. There are many important issues here that are worth highlighting. Maybe include what authors predict will occur in the future in terms of cure, management, screening, etc.
Author Response
24, 40: use “people who use drugs” instead of “people with drug use”
This was changed as recommended above
24: Add “(MSM)” in front of “men who have sex with men”
This was changed as recommended above
27 and beyond: use “HBV” instead of hepatitis B”, as it is already defined in line 21
This was changed as recommended above
40, 54: make sure the reference on outbreaks is up to date as new information is being reported
These data were reviewed and found to be updated
43: “endemicity” instead of “endemnicity”
This was changed as recommended above
52: “occurred in” instead of “identified as”
Changed to occurred as recommended above
91-94: interesting. Consider adding possible explanations for this, e.g. due to blunted immune response, especially if mentioned by authors in references
The following information was added to clarify this point: This is presumably due to a blunted immune response to HAV infected hepatocytes with HIV-associated immunosuppression.
113: add reference to this statement
Reference to the ACIP vaccination recommendations was added
145: “240 million people worldwide are infected with HBV”. Add “chronically” if appropriate
This was changed as recommended above
165 and beyond: nomenclature for HBV antigens and antibodies should be used once spelled out for the first time. Suggest using standard, well accepted nomenclature: HBsAg, HBeAg, anti-HBs, anti-HBc, etc.
This was changed as recommended above
177: Consider adding “Testing for HBV DNA is recommended when isolated anti-HBc is found”. May apply to paragraph 209-213
This statement was added to the end of the paragraph.
197: “HBV antigen”: which one? DNA? Please clarify
This was clarified and changed to HBsAg
200: consider writing this subtitle as in HAV section, line 102 (i.e. choose “vaccination” or “prevention” for both to give consistency)
This section's title was changed to HBV vaccination in people with HIV infection to add consistency as recommended
207: Figure: Box under Chronic HBV infection not clear. Item 3 seems to be missing information: “liver fibrosis…”
The original submission had formatting issues which cut off some of the text. The figure has been re-formatted to include all of the text.
207: Figure: suggest nomenclature as above
This has been changed as recommended
242-243: add “rHBsAg”
This abbreviation was applied
263: define cccDNA
This was defined covalently closed curricular DNA
277: suggest adding much more text to the conclusions as I don’t think that the manuscript is about challenges as described by the authors. There are many important issues here that are worth highlighting. Maybe include what authors predict will occur in the future in terms of cure, management, screening, etc.
The conclusions were expanded to include the following: Challenges in the prevention and treatment of HAV and HBV co-infection among PLWH persist. With regards to HAV, the changing epidemiology of hepatitis A outbreaks highlight the importance of evaluating whether PLWH are immune to HAV when they enter care, and if not, vaccinating them. Development of societal infrastructures that address the underlying root causes of these hepatitis A outbreaks including poverty, homelessness, and substance use are critical. One of the biggest areas of uncertainty is the durability of HAV vaccination among PLWH, and additional data are needed to determine whether periodic monitoring of serologies and/or a booster vaccination beyond the current HAV two-dose vaccine series would be beneficial. With regards to HBV, all PLWH should be screened for evidence chronic infection upon entry into care and, if found to be chronically infected, initiated on treatment with lifelong tenofovir-based, fully suppressive antireotrviral therapy. In cases where tenofovir-based therapy cannot be used, entecavir can be added to a fully suppressive antiretroviral regimen. Ongoing research focusing on virologic and immunologic strategies to eliminate cccDNA and thereby cure HBV is needed, especially for PLWH who are co-infected since this group has worse liver-related outcomes compared to patients with monoinfection. New HBV vaccination strategies that utilize adjuvants such as toll-like receptor agonists are likely to be more effective than the original recombinant surface antigen HBV vaccines in PLWH.
Reviewer 2 Report
Well written, timely, and clinically relevant article. Minor edits/revisions-
Line 42-43: "This is greatest in countries with low hepatitis A endemicity...". Did the authors mean to state "This difference is greatest in countries....)
Line 43: typo "endemnicity"
Line 48-52: "In 22 countries across Europe 4,475 cases of hepatitis A occurred...". Please include over what period were the cases noted.
Line 56: "occur" to be replaced with "occurred"
Line 59: delete '
Line 125-127: Was this difference statistically significant?
Author Response
Line 42-43: "This is greatest in countries with low hepatitis A endemicity...". Did the authors mean to state "This difference is greatest in countries....)
Clarified by re-writing the sentence as: The proportion of PLWH who are seropositive for HAV is greatest in countries with low hepatitis A endemicity and is associated with oral-anal sex, number of sexual partners, older age, and injection drug use.
Line 43: typo "endemnicity"
This typo was corrected
Line 48-52: "In 22 countries across Europe 4,475 cases of hepatitis A occurred...". Please include over what period were the cases noted.
Clarified that the increase in the number of cases has occurred since 2016.
Line 56: "occur" to be replaced with "occurred"
This has been changed to occurred
Line 59: delete '
This was deleted
Line 125-127: Was this difference statistically significant?
This point was clarified that the difference was not statistically significant
Reviewer 3 Report
The authors present a concise yet thorough review of HAV and HBV, focused primarily on the setting of HIV co-infection. I like the focus on vaccination indications, response rate and approaches to re-vaccination. This is particularly key for HBV where much confusion remains among clinicians (in part due to limited data in some areas).
There are a few areas where clarification or additional data/discussion could be added, and I have highlighted those below.
Lines 42-44: It is confusing that the seroprevalence of HAV would be highest in PLWH in non-endemic countries as this sentence suggests. Is “greatest” meant to refer to the difference in HAV seroprevalence between PLWH and those without HIV as opposed to the absolute HAV seroprevalence in PLWH? Maybe change to “This difference in seroprevalence is greatest…” if appropriate.
Lines 91-94. While it is suggested I might add another sentence spelling out that this is presumably due to a blunted immune response to HAV infected hepatocytes with HIV associated immunosuppression (particularly with uncontrolled HIV replication). This would also fit with the next paragraph describing prolonged viremia.
Lines 96-98: might also mention prolonged shedding of HAV in stool which probably has more direct implication for transmission.
Line 166: would suggest specifying HB core total Ab.
Lines 174-177: Would indicate that this is seen more frequently in PLWH, and especially those with HIV/HCV co-infection. Could also consider adding something about HBV DNA testing. While not routinely recommended it may be indicated in certain scenarios like elevated LFTs of unclear cause etc…
Lines 181-182: It may be splitting hairs a bit, tenofovir-based regimens are recommended but that does not necessarily mean 2 anti-HBV agents are needed/recommended. There is essentially no risk of HBV resistance developing to TFV (TDF or TAF). 3TC (and by analogy FTC) regimens where that is the only nuc with HBV activity are not recommended due to high rate of resistance development (as is pointed out later).
I think it would be more correct to say a TFV-based ARV regimen is recommended; and in rare cases where TAF or TDF cannot be used ETV should be added.
Would also consider briefly addressing the small % (5-10%) that do not suppress on TFV-based ART after 3-5 yrs. Data suggests much of this is non-compliance with HIV suppression despite suboptimal compliance while HBV remains detectable. Whether entecavir intensification is clinically beneficial in this setting is unknown.
Figure 1- I note that HCC is defined in the abbreviations but don’t see it in the figure. Was it meant to be #4 in the box after chronic HBV infection as a recommendation for HCC screening?
Author Response
Lines 42-44: It is confusing that the seroprevalence of HAV would be highest in PLWH in non-endemic countries as this sentence suggests. Is “greatest” meant to refer to the difference in HAV seroprevalence between PLWH and those without HIV as opposed to the absolute HAV seroprevalence in PLWH? Maybe change to “This difference in seroprevalence is greatest…” if appropriate.
This point was clarified by specifying that a higher proportion of PLWH are seropositive for HAV in developed countries
Lines 91-94. While it is suggested I might add another sentence spelling out that this is presumably due to a blunted immune response to HAV infected hepatocytes with HIV associated immunosuppression (particularly with uncontrolled HIV replication). This would also fit with the next paragraph describing prolonged viremia.
An additional sentence was added to elucidate this point
Lines 96-98: might also mention prolonged shedding of HAV in stool which probably has more direct implication for transmission.
This point was clarified by stating that prolonged shedding had implications for transmission
Line 166: would suggest specifying HB core total Ab.
This change was made
Lines 174-177: Would indicate that this is seen more frequently in PLWH, and especially those with HIV/HCV co-infection. Could also consider adding something about HBV DNA testing. While not routinely recommended it may be indicated in certain scenarios like elevated LFTs of unclear cause etc…
Recommendations to complete HBV DNA PCR testing was added as recommended.
Lines 181-182: It may be splitting hairs a bit, tenofovir-based regimens are recommended but that does not necessarily mean 2 anti-HBV agents are needed/recommended. There is essentially no risk of HBV resistance developing to TFV (TDF or TAF). 3TC (and by analogy FTC) regimens where that is the only nuc with HBV activity are not recommended due to high rate of resistance development (as is pointed out later).
This point was clarified, emphasizing the importance of a tenofovir-based regimen (instead of 2 HBV active drugs).
I think it would be more correct to say a TFV-based ARV regimen is recommended; and in rare cases where TAF or TDF cannot be used ETV should be added.
This point was changed as above
Would also consider briefly addressing the small % (5-10%) that do not suppress on TFV-based ART after 3-5 yrs. Data suggests much of this is non-compliance with HIV suppression despite suboptimal compliance while HBV remains detectable. Whether entecavir intensification is clinically beneficial in this setting is unknown.
This additional information was added as recommended
Figure 1- I note that HCC is defined in the abbreviations but don’t see it in the figure. Was it meant to be #4 in the box after chronic HBV infection as a recommendation for HCC screening?
This was a formatting issue in which some of the text in the text boxes has been cut off. The figure has been fixed.